# Power supply disruptions deter electric vehicle adoption in cities in China

Yueming (Lucy) Qiu [1,8] ✉, Nana Deng[2,3,8], Bo Wang [3,4] ✉, Xingchi Shen [5], Zhaohua Wang [2,3] ✉, Nathan Hultman [1], Han Shi[3,4], Jie Liu[6] & Yi David Wang[7]

Electrification plays a crucial role in deep decarbonization. However, electrification and power infrastructure can cause mutual challenges. We use nationwide power outage and electric vehicle adoption data in China to provide empirical evidence on how power infrastructure failures can deter electrification. We find that when the number of power outages per district increases by 1 in a given month, the number of new electric vehicles adopted per month decreases by 0.99%. A doubling of power outages in one year on average across the nation can create a depressed adoption rate for up to a decade, implying a decline of more than \$31.3 million per year in carbon reduction benefits from electric vehicle adoptions. This paper adds to the policy discussion of the costs of increased power outages due to extreme weather and natural disasters, and the urgency for policy to address this issue to facilitate wide adoption of electrification.

Low-carbon electrification has gained increasing attention from scholars and policymakers as a central element for reaching deep decarbonization. The recent Intergovernmental Panel on Climate Change (IPCC) report on mitigation of climate change, for example, stresses the urgency to limit global warming to below 1.5° Celsius and emphasizes the importance of electrification in helping achieve this target[1]. Despite the increasing shares of renewable energy in the total energy mix, the end uses in the transportation and building sectors still rely heavily on fossil fuels[2] such as internal combustion engines in gasoline cars and natural gas furnaces and cookstoves in buildings. For example, in 2020, 90% of energy use in the United States (U.S.) transportation sector comes from petroleum products[3]. Electrifying vehicles and buildings while increasing shares of renewables in the power grid can reduce the use of fossil fuels[4–6]. Worldwide, there have been widespread incentives (e.g., tax credits, direct rebates, and low-cost financing)[7] for electrification that encourage the adoption of technologies such as electric vehicles and heat pumps.

However, a core premise of electrification is that there would be a steady electricity supply to power electric vehicles or electric heating and cooling appliances for buildings. Nevertheless, future acceleration of electrification can potentially face pressure in the complex energy and social systems. The pressure is twofold. On the one hand, with the targets of increasing renewable energy to power the electric grid by many countries, there has been a concern over whether such high penetration of renewable energy imposes challenges to a stable electricity supply[8,9] due to intermittency issues. On the other hand, the electrification process will also make the electricity demand much higher, further adding challenges to the electric grid resilience[10]. If there is an unpredictable surge in electricity demand during system peak hours due to electrification that the generating capacity cannot meet, power outages can happen[11]. This study provides empirical evidence using nationwide data in China on the negative impact of power outages on electric vehicle (EV) adoption, an important way of electrification, from November 2019–September 2021.

[1]School of Public Policy, University of Maryland at College Park, College Park, MD 20742, USA. [2]School of Economics, Beijing Institute of Technology, Beijing 100081, China. [3]Digital Economy and Policy Intelligentization Key Laboratory of Ministry of Industry and Information Technology, Beijing 100081, China. [4]School of Management, Beijing Institute of Technology, Beijing 100081, China. [5]Yale School of the Environment, Yale University, New Haven, CT 06511, USA. [6]Institute of Technology for Carbon Neutrality, Shenzhen Institute of Advanced Technology, Chinese Academy of Sciences, Shenzhen 518000, China. [7]Department of Economics, Virginia Tech, Blacksburg, VA 24060, USA. [8]These authors contributed equally: Yueming (Lucy) Qiu, Nana Deng. ✉e-mail: yqiu16@umd.edu; 51022080@qq.com; wangzhaohua@bit.edu.cn

Even without climate mitigation policies, the challenges of power supply disruptions in the electrification process are pressing. This is because insecure electricity supplies in many parts of the world are almost as likely to stem from non-climate policy-related failures in electricity system planning and/or the impacts of unmitigated climate change itself[12–14], which places significant stress on existing electricity infrastructure. For example, in this study context, the key challenges to power infrastructure stability come from the sharp increase in electricity demand due to rapid urbanization as well as extreme weather events[15].

Countries such as China and the U.S. have experienced increasing pressure from power outages due to extreme weather events such as winter storms, heat waves, hurricanes, and wildfires. The severe power outages in California, Texas, and Louisiana in 2020 and 2021 caused by extreme weather conditions have sparked a discussion on upgrading energy infrastructure and improving cities' power resilience to better cope with extreme weather and related disasters. The recent power outages in California, Texas, and Louisiana caused by extreme weather or disasters could cost $ 2 billion[16], $80–$130 billion[17], and $31–$44 billion[18] respectively. In 2021, there were unprecedented large-scale power outages in China. The power cuts and blackouts in September and October of 2021 have slowed or closed factories across China and left millions of homes hit by power cuts[19]. These blackouts were mainly driven by a surge in demand for Chinese goods after the re-opening of the global economy after the pandemic, as well as the surge in coal prices, making coal-fired power plants unwilling to operate at a loss. In addition, there was a temporary supply shortage due to flooding events in Sep 2021 that caused coal mines to suspend operations[20,21]. There were also large-scale power outages in southern China during winter in Dec 2020 and Jan 2021, largely due to extreme cold weather and a surge in electricity demand for heating purposes[22,23].

Despite the increasing severity of power outages, the impact and implications of power outages on the low-carbon electrification process have not yet been studied empirically. Existing studies mainly focus on another type of electrification in developing countries, namely rural electricity access, and they have found that power outages will reduce the expected benefits from rural electrification[24,25]. This study examines the impact of power outages on the electrification of the transportation sector. If there are frequent or extended power disruptions, consumers relying on electric vehicles may have difficulty driving especially for essential needs such as purchasing food or going to the hospital. There has been abundant literature identifying the factors influencing the adoption of EVs, such as range anxiety[26,27], availability of EV charging stations[28,29], environmental awareness[30,31], and income[32,33], while a recent statistical study shows no causal relation between EV charging stations and EV purchases[34]. Among these factors, range anxiety and availability of charging stations indirectly imply that a steady supply of electricity to EVs is a critical factor. However, there is a lack of empirical evidence on the impact of power supply disruptions on EV adoption. Such empirical evidence is important for policymakers to invest adequately in grid infrastructure to ensure electrification's steady development and diffusion. The analysis of this paper demonstrates the importance of this factor for establishing policies and calculating the costs and benefits of grid infrastructure investment—for both the direct impact on EV adoption and the potential consequences on delaying carbon emissions reductions. Mildenberger et al.[14] conduct a survey among residents in California and find that power outage-exposed survey respondents were 7 percentage points less likely to report that they intended to buy an EV as the next car. This survey study helps support this study's hypothesis, though it relies on consumers' reported plans, which may deviate from their actual purchase decisions. Empirical research using actual consumers' EV purchase data is needed. The nationwide empirical analysis of cities in China provides further quantitative evidence based on actual EV sales data to support this study's hypothesis.

This study provides three main contributions. First, this study compiles the nationwide high-frequency point-level power outage database of cities in China spanning from November 2019 to September 2021. This database records detailed information for every power outage, such as the start and end times, addresses and areas affected, and causes of power outages. The high-resolution data makes it possible to link the variation of power outages to other important outcomes in the energy systems such as EV adoption. Second, this study provides empirical evidence on how power system failures can deter electrification using EV adoption as an example. As a placebo test, this study also finds that power outages cause an increase in the adoption of non-EVs, possibly due to a substitution effect between the adoption of EVs and non-EVs. Lastly, this study estimates the reduced carbon emissions benefits from the delay in EV adoptions due to power outages. This study estimates that a doubling of power outages in 1 year on average across the nation can create a depressed adoption rate for up to a decade, implying a decline of >$ 31.3 million/year U.S. dollars of carbon reduction benefits from EV adoptions. Assuming that this delay is a decade, the discounted total reduced benefits will equal $ 254 million U.S. dollars. One important caveat of the extrapolation of the short-term impact to longer-term carbon reduction impact is that it requires assumptions such as the stability through time in the policy environment, consumer preferences, and the social cost of carbon. Please note that this is only the impact of a 1-year shock of doubling power outages. More severe power outages in more years will generate much higher damages. This will pose challenges to limiting global warming to under 1.5 °C.

## Results
### Recent large-scale power outages in China
The point-level high-frequency power outage data spans from November 2019–September 2021. This study obtained this data by scraping the daily power failure data published on the official website of each city government. During this timeframe, there were two time periods with large-scale power outages happening across provinces. This study defined large-scale power outages as situations where multiple provinces experience extended electricity shortages and take power rationing measures around the same time. From Dec 2020 to Jan 2021, the unprecedented low temperatures in China caused a huge surge in demand for heating that increased electricity demand[35]. The sharp increase in electricity demand, especially in southern provinces that do not have central heating in place, could not be met by existing generating capacity, resulting in power outages. The government also issued mandatory notices to communities and businesses to restrict electricity usage during these times[36]. Based on the data, from Dec 2020 to Jan 2021 the most impacted provinces in terms of the average district-level (a city can have multiple districts) number of power outages are Zhejiang (642 counts of power outages/district in 2 months), Jiangsu (619), and Anhui (498); the most impacted provinces in terms of the average district-level hours with power outages are Hunan (4719 h/district in 2 months), Hubei (3909), Sichuan (3732), Jiangsu (3668), and Guangdong (3629). In Sep and Oct of 2021, an even larger scale of power outages happened, not just in southern China but across the whole nation. On the demand side, the recovery of the global economy from COVID-19 significantly increased the demand for Chinese goods, which needed electricity to produce. On the supply side, the surge in coal prices caused coal-fired power plants to shut down operations to avoid selling electricity at a loss[37]. In addition, there was a temporary supply shortage due to flooding events in Sep 2021 that caused coal mines to suspend operations[20,21]. During these 2 months, many factories in China were impacted and closed. For example, the aluminum industry in Yunnan reduced production by 11–24% due to the power outages[38]. Based on the data, in Sep 2021 the most impacted provinces in terms of the number of power outages are Jiangsu (786 counts/district in 1 month), Zhejiang (308), and Anhui

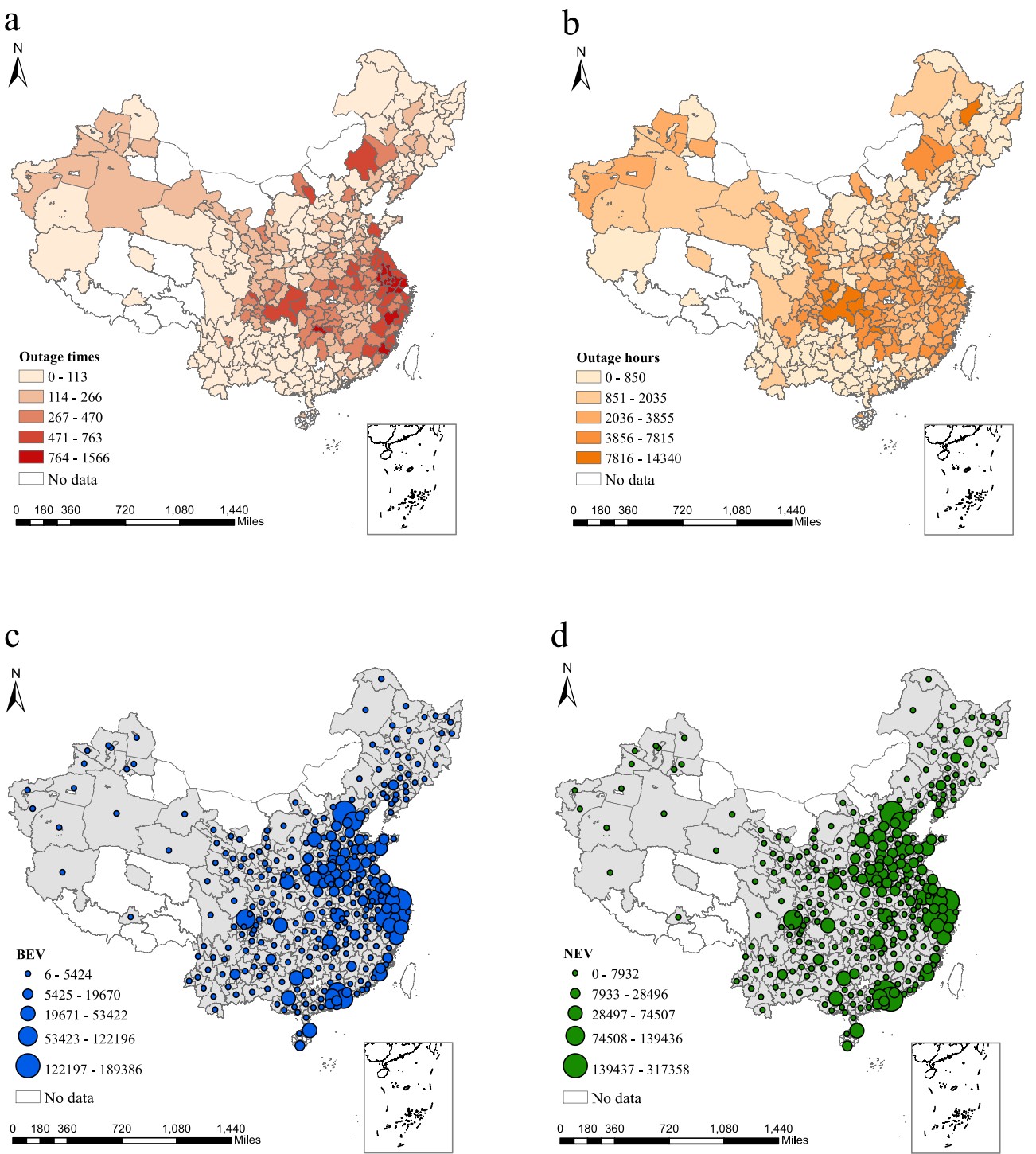

**Fig. 1 | Geographic distributions of power outage conditions and EV sales across 320 cities in China, 2019.11-2021.09. a** shows the total number of power outages during the study period per district of a city. A darker red color indicates a higher frequency of outage times. **b** shows the total number of hours with power outages at the district level in a city. A darker shade of orange indicates a longer cumulative duration of power outages in the city. **c** shows the total battery electric vehicles (BEVs) sales at the city level. The gray shades denote cities with available sales data, while larger blue circles indicate higher BEV sales in the city during the sample period. **d** shows the total sales of new energy vehicles (NEVs) from November 2019 –September 2021. Here, the total sales of NEVs are equal to the combined sales of both BEVs and plug-in hybrid electric vehicles (PHEVs). Larger green circles indicate higher NEV sales in the city. The blank areas represent cities with no data available. Source data for this figure are available on GitHub.

(279); the most impacted provinces in terms of the hours with power outages are Henan (9486 h/district), Jiangsu (5121), Hunan (3507), and Sichuan (3427).

Figure 1 shows the distribution of the average power outage situation and new EV sales from Nov 2019–Sep 2021. The data of this study shows regions with intense power outages during this time frame, including parts of Shandong, Jiangsu, Zhejiang, Fujian, Hubei, and Hunan provinces, as well as Shanghai, Chongqing, and the Inner Mongolia Autonomous Region. Some districts experienced >1500 counts of power outages and >14,000 h during the 669 days, which is

equal to an average of 2.24 power outages and 21 h with no power per day. Please note that the hours of power outages per district of a given time frame calculated in the dataset are the summation of hours that happened in all locations within a district. Please see details of this calculation method in the Methods section. The lack of a steady electricity supply in these severely impacted cities could exacerbate the charging availability concerns of potential EV buyers and, thus, lower EV adoption. Figure 1 only shows a descriptive distribution of power outages and EV sales. In the next section, this study uses econometric models to tease out the confounding factors and identify the relationship between power outages and EV sales.

## EV adoption reduced due to power outages

This study constructs a monthly panel dataset at the individual city level, using the number of EVs that receive new automobile insurance each month as a proxy for new EV sales. The key explanatory variable is the power failure situation of each city. This study uses fixed-effects panel regression to regress the monthly EV sales at the city level on lagged city-level power outages (1- and 2-month lags), while controlling for factors such as per capita GDP and the number of EV charging stations. The rich fixed effects can further control for unobservable time-variant confounding variables at the city level, such as the incentives and policies in place for adopting EVs, population, consumer environmental awareness, and area of paved roads as well as seasonal holiday, climate, temperature (which influences battery performance), and business cycles. Supplementary Table 1 shows the summary statistics of the key variables in the regression. Details of the data and models can be found in the Methods section.

Table 1 shows the regression results. In all model specifications, power outages have a statistically significant and negative impact on EV adoption. Column (1–2) shows that when the number of power outages per district increases by 1 of a given month, the number of new EVs, including battery electric vehicles (BEVs) and plug-in hybrid electric vehicles (PHEVs), adopted per month decreases by 1.1% (1-month lag effect) and 0.88% (2-month lag effect) in a city. A formal statistical test (Supplementary Table 22) shows that the coefficients for PHEV versus BEV do not differ statistically[39]. Based on the percentage decrease estimated from the models for BEVs (0.92%) and PHEVs (1.4%), as well as the sales volume of BEVs (2.734 million in 2021) and PHEVs (0.6 million in 2021), this study calculates that in terms of the decrease in the number of sales, power outages have a greater impact on BEVs (25152 cars) than PHEVs (8400 cars).

Column (3–4) shows that when the average number of hours with power outages at the district level of a given month increases by 1 h, the number of new EVs adopted per month decreases by 0.024% (1-month lag effect) and 0.021% (2-month lag effect) in a city. In a robustness check, this study adds the number of EV charging stations as an additional control variable. Supplementary Table 2 shows that with the number of EV charging stations added as a control variable, the impact of power outages on EV adoption remains consistent with the main results. The results of the negative impact of unstable electricity supply on EV adoption are related to the links between counterfactual supply stability facing consumers under the combustion-engine car, namely the links between gas prices, stability of petroleum supply, and personal car purchases. Studies have found that the increase in gas prices or the disruption in petroleum supply reduces combustion-engine car purchases[40,41].

## Robustness checks

In order to further strengthen the main conclusions, this study conducts a placebo test to estimate whether power outages have a negative impact on traditional non-EV vehicles that do not need to plug into the power grid. If there were a negative impact on the non-EV vehicles, then the earlier estimated negative impacts on EVs could be due to other unobserved factors rather than power outages. Table 2 shows

**Table 1 | Regression results showing the negative impact of power outages on EV adoption**

| | lnNEV (BEV + PHEV) | | | | lnBEV | | | | lnPHEV | | | |
|---|---|---|---|---|---|---|---|---|---|---|---|---|
| | (1) | (2) | (3) | (4) | (5) | (6) | (7) | (8) | (9) | (10) | (11) | (12) |
| L1. (outage times) | -0.011*** (0.001) | -0.0053*** (0.001) | | | -0.0092*** (0.001) | -0.0039*** (0.0009) | | | -0.014*** (0.003) | -0.0016*** (0.003) | | |
| L2. (outage times) | | -0.0088*** (0.001) | | | | -0.0076*** (0.0012) | | | | -0.023*** (0.003) | | |
| L1. (outage hours) | | | -0.00024** (0.0001) | -0.00015 (0.00008) | | | -0.00020** (0.00009) | -0.00011 (0.00007) | | | -0.00027* (0.0001) | -0.00031* (0.00016) |
| L2. (outage hours) | | | | -0.00021*** (0.00007) | | | | -0.00019*** (0.00007) | | | | -0.00050*** (0.0001) |
| ln GDP | -0.042 (0.027) | -0.048** (0.023) | -0.043 (0.027) | -0.048** (0.023) | -0.024 (0.026) | -0.026 (0.021) | -0.025 (0.026) | -0.026 (0.022) | -0.047 (0.040) | -0.074 (0.046) | -0.049 (0.042) | -0.075 (0.047) |
| Constant | 4.63*** (0.22) | 4.94*** (0.19) | 4.55*** (0.22) | 4.82*** (0.19) | 4.31*** (0.21) | 4.57*** (0.18) | 4.25*** (0.22) | 4.47*** (0.18) | 2.48*** (0.34) | 3.14*** (0.39) | 2.37*** (0.34) | 2.79*** (0.39) |
| R-squared | 0.96 | 0.98 | 0.96 | 0.98 | 0.97 | 0.98 | 0.96 | 0.98 | 0.90 | 0.90 | 0.90 | 0.90 |
| Year*City FE | YES | YES | YES | YES | YES | YES | YES | YES | YES | YES | YES | YES |
| Month*City FE | YES | YES | YES | YES | YES | YES | YES | YES | YES | YES | YES | YES |
| Observations | 4700 | 4090 | 4700 | 4090 | 4700 | 4090 | 4700 | 4090 | 4700 | 4090 | 4700 | 4090 |
| Number of city | 301 | 298 | 301 | 298 | 301 | 298 | 301 | 298 | 301 | 298 | 301 | 298 |

This table reports the estimated coefficients and cluster-robust standard errors (in parentheses). The dependent variable in Columns (1–4) is the log of monthly sales for new energy vehicles (NEVs); in Columns (5–8), it is the log of monthly sales for battery electric vehicles (BEVs); and in Columns (9–12), it is the log of monthly sales for plug-in hybrid electric vehicles (PHEVs). Here, the total sales of NEVs equal the combined sales of both BEVs and PHEVs. L1 means 1-month lag; L2 means 2-month lag. Standard errors in parentheses are clustered to the city level. *$P < 0.1$, **$P < 0.05$, ***$P < 0.01$. R-squared denotes the goodness-of-fit of the regressions.

**Table 2 | Regression results showing the non-negative impact of power outages on non-EV adoption**

| ln non-EV (Gasoline+Diesel+Compressed natural gas+Ethanol) | | |
|---|---|---|
| L1.Outage times | 0.038*** | |
| | (0.005) | |
| L1.Outage hours | | 0.0007*** |
| | | (0.0003) |
| lnGDP | 0.086 | 0.091 |
| | (0.088) | (0.09) |
| Constant | 3.84*** | 4.13*** |
| | (0.73) | (0.75) |
| Year*City FE | YES | YES |
| Month*City FE | YES | YES |
| Observations | 4700 | 4700 |
| Number of city | 301 | 301 |
| R-squared | 0.67 | 0.65 |

This table reports the estimated coefficients and cluster-robust standard errors (in parentheses). The dependent variable is the log of monthly sales for non-energy vehicles (non-EVs), which include cars powered by gasoline, diesel, compressed natural gas, and ethanol. L1. means 1-month lag. Standard errors in parentheses are clustered to the city level. *$P < 0.1$, **$P < 0.05$, ***$P < 0.01$. $R$-squared denotes the goodness-of-fit of the regressions.

that power outages do not reduce the sales of non-EV vehicles, including cars that run on gasoline, diesel, compressed natural gas, and ethanol. The non-negative impact on non-EV sales indicates that the estimated impact on EVs is not likely due to confounding factors that can cause lower demand for cars in general, such as road infrastructure conditions and travel behavior patterns. This helps justify the main results of this study. In fact, power outages increase the sales of these non-EV vehicles.

In another robustness check, this study adds a control variable that measures local governments' mandatory notices to restrict power usage among communities and businesses during power outages. This study collects such information from official government documents and notices. This study generates a dummy variable that is equal to 1 if a region in a given month issued such a mandatory order. Results are shown in Supplementary Table 16. The results show that the coefficients for the power outage variables remain similar to the main model results, confirming the robustness of the results of this study. In addition, mandatory orders have a negative impact on EV sales. We also incorporated more socio-economic variables into the model, including population, area of paved roads, coal gas supply, petroleum gas supply, educational level (measured by the number of teachers in ordinary higher education institutions), access to public transport (measured by the number of public buses) and charging infrastructure reliability (measured by the number of EV charging stations). The regression results are shown in Supplementary Table 23. The coefficients for the power outage variable remain significantly negative, confirming the robustness of the results of this study.

There is a possible simultaneous relationship between EV purchases and power outages. Rapid increases in EV purchases likely cause electricity consumption to rise. And such increases in consumption can increase the likelihood of power outages, through insufficient supply or deficiencies in the grid. If there is a simultaneous relationship between EV purchases and power outages, the single equation approach will bias estimates for the regression coefficients. To further address the potential endogeneity issues, this study conducts an instrumental variable (IV) approach (details of the IV methods can be found in Methods), using monthly extreme temperature days as the IV, where extreme temperature is defined as daily maximum temperature > 89.6 °F or daily minimum temperature < 32 °F. The IV results in Supplementary Table 18 and Supplementary Table 19 (which estimate

the local average treatment effects of the impact of power outages that are influenced by extreme temperature) show that the coefficients for both the number of power outages and the hours of power outages remain statistically significant and negative, confirming the causal relationship between power outages and EV sales. This implies the robustness of the main models and further confirms the causality identified by this paper from the main models. In order to further justify the main empirical findings, this paper conducted a survey of 890 consumers across 31 provinces in China in Oct 2022, and the survey results indicated that increasing power outages will influence the purchase intention of potential EV buyers (Details of the survey can be found in the Methods section).

Disruptions in the supply chain and other impacts of COVID-19 may also have a negative impact on EV sales. Chinese government enforced lockdown measures in cities experiencing the pandemic. Such lockdown and stay-at-home policies reduced economic activities, disrupted supply chains, decreased personal mobility, and banned many forms of travel[42–44]. To account for these factors, this paper uses a city lockdown dummy variable to measure the immediate impact of COVID-19. It uses the lagged term of the lockdown variable to measure the impact of disruptions in the supply chain. Due to data limitations, this study was not able to find city-level data on supply chain disruptions and thus use the lagged term of COVID-19 lockdown measures as a proxy for supply chain impact, since the supply chain responses may lag the issuance of the lockdown measures. The results of the models adding these two variables are shown in Supplementary Table 17. The results are consistent with the main conclusions and show that, conditional on the shocks of COVID-19 restrictions and related disruptions in supply chains, power outages still have a statistically significant and negative impact on EV sales. Results also show that COVID-19 lockdown policies and the associated disruptions in the supply chain had a negative impact on EV sales. As an additional measure, we excluded data from cities that were significantly impacted by COVID-19. We identified cities listed as medium or high-risk areas during the sample periods as regions severely affected by the pandemic. The regression results are shown in Supplementary Table 24. The results are still consistent with the main conclusions.

Power outages may trigger general economic shifts that change consumer behaviors. This study runs a regression to examine the relationship between power outages and GDP/ industrial value added. Supplementary Table 21 shows that power outages do not cause lower GDP or industrial value added.

## The decline in carbon emissions reductions
From Table 1, this study takes the average of the coefficients for the 1-month lag and the 2-month lag as the effect of the power outage on EV adoption, which is equal to $(0.011 + 0.0088)/2 = 0.0099$. This is a reduction of 0.99% in EV adoption if there is one more power outage per district in a city for a given month. At the average new monthly EV sales (including BEV and PHEV) of a city across the nation at 405 cars per month, 1 more power outage per district per month can decrease about 4.01 EVs (0.0099*405) in a city per month. Considering the nationwide impact, with 310 cities in total in the sample, 1 more power outage per district per month can decrease 1243 EVs (0.0099 * 405 * 310) across China in a month. Using the estimates from Peng et al.[45] and Caixin[46], the monetized carbon reduction benefit of an EV in China is about $ 210 (US dollars) per year (see details in Methods), which is calculated based on the social cost of carbon and the annual reduction of $CO_2$ from an average EV. If, on average, every month in a year has 1 more power outage per district in the whole nation, this 12 month period may cause a reduction of $ 3.13 million (210 * 1243 * 12) U.S. dollars of climate benefits from EV adoption in cities.

The large-scale power outages in 2020 and 2021 in China delayed China's EV adoption significantly. During these 2 months, the top three

**a**

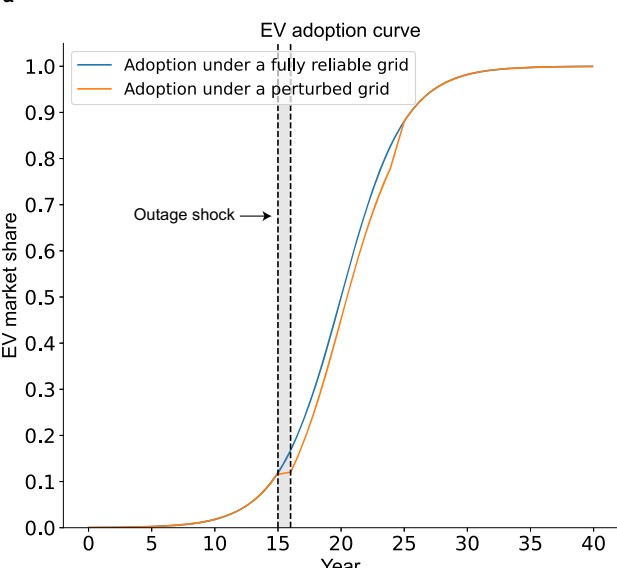

**b**

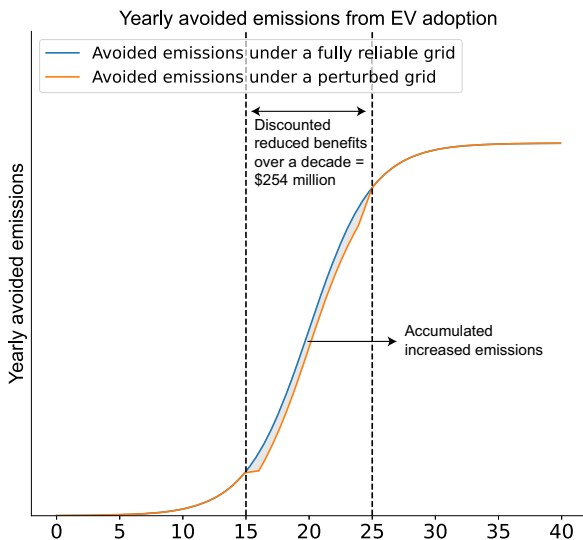

**Fig. 2 | Conceptual framework to demonstrate the impact of a one-year power outage shock on electric vehicle (EV) adoption and carbon emissions reductions. a** depicts the yearly EV adoption, with orange line representing the EV market share under a perturbed grid and blue line representing the EV market share under a fully reliable grid. The gray shadow in (**a**) indicates the outage shock. The cumulative adoption curve is constructed based on Grubb et al.[86] and Rogers et al.[87], which show that the diffusion curve of EVs should follow an S-curve trajectory. **b** depicts the yearly avoided emissions from EV adoption, with orange line representing the avoided emissions under a perturbed grid and blue line representing the avoided emissions under a fully reliable grid. The shaded area in (**b**) is the cumulative increased carbon emissions due to the reduced EV adoption. Key assumptions in this figure: (1) there are no other policy interventions or other shocks after the power outage increase in that year; (2) the delay is 10 years if consumers use their newly purchased fossil fuel vehicles for 10 years; (3) the carbon reduction amount of an EV stays constant over time; (4) the social cost of carbon stays the same. Source data for this figure are available on GitHub.

provinces with severe power outages are Zhejiang (642 counts of power outages/district), Jiangsu (619), and Anhui (498). These are >24 times the average power outage numbers per district in 2 months across the nation (20 counts in 2 months). Assuming that the power outages double in a given year on average across China (meaning an increase from 10 to 20 on average per district per month), this will cause a decline of $ 31.3 million/year (10 * 0.99% *405 * 310 * 210 * 12) U.S. dollar of carbon reductions benefits. Assuming that this delay is a decade, then the discounted total reduced benefits will be equal to $ 254 million U.S. dollars using a 5% discount rate (typical climate models use discount rates between 2–7%[47]).

The $31.3 million/year is only the carbon reduction benefits with the current fuel mix of the electric grid across China. If this study also considers the air pollution benefits from reduced pollutants such as particulate matter, sulfur dioxide, and nitrogen dioxide, as well as the associated health benefits, the total reduction in carbon, air quality, and health benefits from reduced EV adoption due to power outages will be much higher. Even after considering these other benefits of EVs, the estimated impact is still likely to be the lower bound. First, currently, this study finds that one outage per city per month can decrease EV adoption by about 1%. On average, new monthly EV sales (including BEV and PHEV) in a city across the nation are 405 cars per month. This number will be much larger for the next decade, according to the S-shape curve of EV diffusion and carbon-neutrality commitment. Thus, in the future, the impact of the outage on the number of EV adoption will be much larger. Second, the social cost of carbon will be larger as the year progresses. According to EPA's estimates on SSC[48], the SSC will be larger in future years (after adjusting for inflation). Thus, the carbon reduction benefits from EVs will be even larger in the near future. Third, as the future electric grid becomes cleaner, the amount of carbon emissions avoided by an EV will be greater.

In order to provide an estimate of the upper bound of reduced benefits due to delayed EV adoption resulting from power outages, this study uses the estimates of the study by Peng et al.[45] on the economic

benefits of combining health benefits and GHG emission reductions due to the maximum penetration of alternate energy vehicles with a high fraction of renewable electricity. This study found that the upper bound of decline in benefits from reduced EV adoptions in cities due to power outages is equal to $ 911 million/year U.S. dollars (Detailed calculation can be found in Methods).

This decline in carbon reduction benefits caused by power outages can be as long as a decade. Every year, the new consumer demand for replacing old cars and buying new vehicles is released. Consumers face different choices between EVs and non-EVs. If some consumers buy fossil fuel vehicles because of power outages, they will use their fossil fuel vehicles for at least several years or even a decade. Thus, the delay of environmental and carbon benefits due to the drop in EV adoption caused by power outages can be several years to a decade. Figure 2 is a conceptual framework to illustrate the impact on EV diffusion and carbon emissions reductions from a one-time shock, namely the doubling of power outage frequency in one year. Please note that Fig. 2 is only a conceptual framework and does not intend to simulate the actual EV diffusion curve in China, which requires more data beyond the scope of this paper. Figure 2a shows that after the one-year shock, the cumulative EV diffusion in the next 10 years will be lower at an almost constant rate compared to the counterfactual scenario (if there were no increase in power outages). This is because a portion of consumers who experienced the increased power outages of that year purchased fossil fuel vehicles, and these vehicles won't get off the road until a decade later. Panel (b) shows the corresponding reduced carbon benefits due to the yearly lower diffusion rate. The discounted total reduced benefit is equal to $ 254 million U.S. dollars, calculated using the $ 31.3 million/year carbon reduction benefits.

## Heterogeneous impact of power outages on EV adoption

This study examines how the impact of power outages on EV adoption differs by vehicle type, region, and GDP levels. In terms of vehicle type,

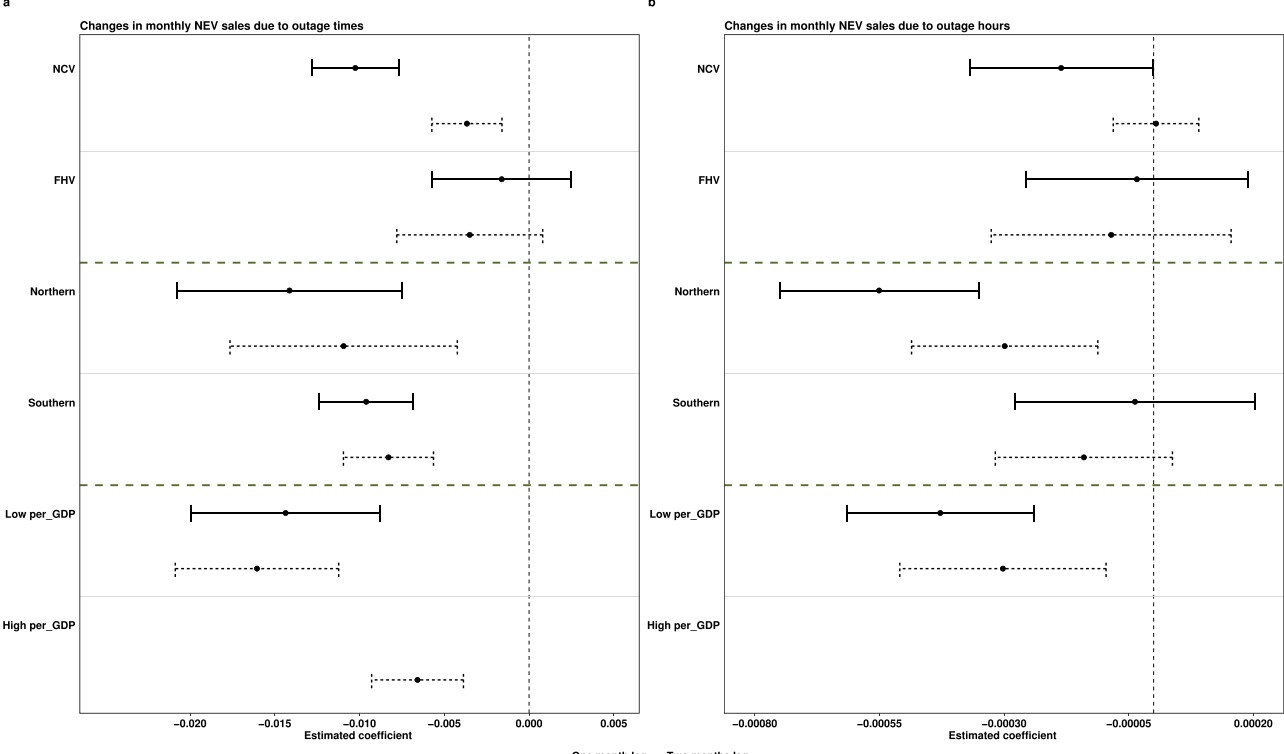

**Fig. 3 | Heterogeneous impact by vehicle type, region, and GDP.** This figure plots the coefficients of the power outage variables in the regression models specified in Supplementary Tables 3, 4, 5 and 6. The centers of the error bars are the values of the coefficients. The solid horizontal lines are the 95% confidence intervals for analyses using a 1-month lag in power outages and dashed horizontal lines correspond to analyses using a 2-month lag in power outages. **a** shows the coefficients for the number of power outages; **b** shows the coefficients for the hours with power outages. NCV is non-commercial private vehicle; FHV is for-hire vehicle. The northern versus southern provinces are divided by the Huai River. High GDP indicates provinces with a per capita annual GDP >60,000 RMB; low GDP indicates provinces with a per capita annual GDP <60,000 RMB. For the analysis of NCV and FHV, the data include battery electric vehicles (BEVs), plug-in hybrid electric vehicles (PHEVs), and non-plug-in hybrid electric vehicles due to data limitations. For the analyses by region and by GDP, the data only include BEV and PHEV. The detailed regression results can be found in Supplementary Tables 3,4,5&6. The relatively insignificant results of the coefficients when using the hours of power outages could be due to less variation in power outage hours compared to the number of power outages. Source data for this figure are available on GitHub.

this study compares the impact on non-commercial vehicles (NCV) and for-hire vehicles (FHV). In the categorization, FHV includes taxis, cars owned by rental companies, cars owned by drivers of car-sharing companies, and cars owned by government agencies or corporations. The results in Fig. 3 show that power outages do not have a statistically significant impact on the adoption of FHVs, which could be due to the fact that for-hire vehicles are mainly purchased by government or private companies to demonstrate the commitment to electrification. Their procurement decision is more influenced by top-down policies and, thus, less influenced by power outages. In addition, larger organizations may plan their fleet procurement with a longer time horizon and may even have access to or could invest in ensuring a more stable electricity supply, e.g., backup generators if needed. There have been various policies and measures requiring the government to purchase and use electric vehicles for their official vehicles[49,50]. Similarly, the central and local governments have stated in their policy documents that the shares of clean energy vehicles used for industries such as taxis, ride-hailing services, urban logistics and freight distribution, and express delivery need to be improved[51,52]. As a result, there is policy pressure for companies and government agencies to increase the adoption of electric FHVs, and thus, their decisions will not be sensitive to outages. However, the effectiveness and sustainability of these policies may be challenged by infrastructure limitations, notably the reliability of the electricity supply. Given the critical role of a reliable electricity supply in the successful adoption of EVs, frequent or prolonged power outages could necessitate a re-evaluation of existing

policies. Acknowledging and addressing these challenges through adaptive policies could pave the way for more resilient and effective electrification efforts in the transportation sector.

The number of power outages has a negative and significant impact on the adoption of NCVs. Note that due to data limitations, the categorization of NCV versus FHV cannot distinguish between PHEV and non-plug-in hybrid vehicles, so this analysis includes non-plug-in EVs, which should not be influenced by power outages due to not needing to plug into the grid. In terms of regions, cities in northern provinces experience a more negative impact on EV adoption from power outages than those in southern provinces. The difference between northern and southern regions could be due to GDP differences, where southern provinces, on average, have a higher GDP. Similarly, Fig. 3 shows that cities with lower GDP experience a more negative impact compared to those with higher GDP. Consumers in lower GDP regions might have lower disposable income and are thus more sensitive to any negative aspects of EVs. In addition, EVs are less popular in northern cities due to the colder climate, which makes charging difficult[53]. This can also explain why power outages have a more negative impact in the north.

This study tests whether the differences are statistically significant by adding interaction terms between the power outage variable and category indicator variables in the regression analyses (see details in methods). Results in Supplementary Table 7 show the differences between regions and GDP levels are statistically significant in some model specifications.

## Discussion

This study uses nationwide power outage and EV sales data of cities in China to provide empirical evidence on how power infrastructure failures can deter the adoption of electric vehicles. This study creates high-resolution geo-reference point-level power outage data in China. This study finds that when the number of power outages per district increases by 1 of a given month, the number of new EVs, including BEVs and PHEVs, adopted per month decreases by 0.99% in a city. Such a negative impact is economically significant. A doubling of power outages in 1 year on average across the nation can create a depressed adoption rate for up to a decade, implying a decline of >$ 31.3 million/year U.S. dollars in carbon reduction benefits from EV adoptions in cities. Assuming that this delay is a decade, then the discounted total reduced benefits will be equal to $ 254 million U.S. dollars. Please note that this is only the impact of a 1-year shock of doubling power outages. More severe power outages in more years will generate much greater damage. Policymakers need to estimate the costs of power outages in order to adopt efficient policies to improve grid resilience while also meeting the carbon emissions reduction targets. This paper adds to the discussion of the costs of power outages and potential challenges to wide electrification by providing empirical evidence on how power outages can deter EV adoption and the associated carbon emissions reductions in China.

These results have important implications for the electrification process, not only for China and the U.S. but also for any country or region that is seeking to accelerate EV deployment under a rapidly decarbonizing electricity supply and distribution infrastructure, in a climate with potentially increasing hazards. Major countries around the world are seeking to facilitate more rapid electrification through policy instruments and technological breakthroughs[54]. With elevated electricity demand and the associated uncertain impact on the electric grid from electrification, together with increased natural disasters and extreme weather events, as well as increased penetration of renewable energy in the electricity generation mix, power outages could cause more challenges in the near future. The results highlight that uncertainty in the stability and reliability of the power infrastructure could delay this critical low-carbon electrification process, causing significant delays in the environmental, health, and carbon reduction benefits. The IPCC report has documented the urgency for radical reductions in carbon emissions in order to limit global warming to 1.5°C[1]. The delays in the electrification process due to power infrastructure failures will add a significant challenge to tackling global warming. The non-climate drivers of power outages indicate the scenarios of the potential impact of power outages on electrification progress under policy counterfactuals. In other words, even without climate-mitigation challenges, the aging power infrastructure and extreme weather events will cause increasing pressure on power supply stability. As a result, the policy implications of strengthening the power infrastructure are even more important to consider.

Several measures and policy approaches can be adopted to address this challenge from power outages and power resilience during the low-carbon electrification process. First, new investment can be directed to increase the resilience of the power grid, especially in regions and countries where the power grid is relatively old, or there is a lack of adequate power infrastructure, such as in the United States[55,56] and many developing countries[57,58]. Second, the generation capacity needs to be re-optimized stage-by-stage to meet the increased electricity demand due to electrification. This needs advanced power grid operations and balancing as well as better forecasting and understanding of the new power load profiles when higher penetration of electrification happens[11]. Third, smarter and more advanced demand-side management is needed that can flexibly utilize electric vehicles for load controls. For example, vehicle-to-X (V2X) can connect EVs with buildings and the electric grid to better achieve load-flattening purposes. Lastly, options for storage, including but not limited to residential and fleet battery storage, should be further developed to improve resilience in case of severe power outages[59], which can relieve the anxiety faced by consumers to relieve the potential impact of power outages on the electrification process.

In addition, behavioral interventions have been demonstrated helpful in reducing peak demand and flattening the load curves, which will improve the grid stability and lower the probability of power outages. For example, studies have shown that time-of-use pricing, which imposes higher electricity prices during peak hours with higher electricity demand and lower electricity prices during off-peak hours, can effectively shift part of consumers' electricity demand from peak hours to off-peak hours and thus help improve the power grid stability[60-63]. There are also other pricing interventions, such as critical peak pricing, critical peak rebate, and real-time pricing, that can effectively help flatten the load curve, reduce demand during peak hours, and lower the likelihood of power outages[64-66]. Other behavioral interventions such as moral persuasion and peer effects via home energy reports, or utility messaging to the consumers have also been shown to be effective at conserving electricity and thus reduce the pressure on the electric grid[65,67]. Chinese governments and electric utilities have implemented a variety of demand mitigation policies to help balance the super-peak demand in the summer and winter seasons. For example, electric utilities invite certain private companies and residential households to participate in a demand-response program during peak periods[68]. Invitees are asked to lower their electricity consumption for certain hours and receive a certain electricity subsidy in return. In addition, as of 2021, 29 provinces in China have introduced time-of-use rate plans, which charge higher electric prices during peak periods to help shave peak demand[69]. In recent years, more provinces (e.g., Zhejiang and Xinjiang) have increased the gap between the peak price and the off-peak price for commercial and industrial consumers to enhance the impact of rate policy on demand mitigation[70,71].

## Methods

### Ethics declarations

This research complies with relevant ethical regulations. University of Maryland College Park (UMCP) Institutional Review Board (IRB) approved the survey study protocol. The IRBNet ID for this study is 2209723-1.

### Data

This study constructs a monthly panel dataset spanning from November 2019 to September 2021 at the individual city level. Although the power outage data is high frequency and at the geo-reference point level, the EV sales data is only at the city level. This study uses the number of EVs getting automobile insurance each month as a proxy for new EV sales. This study obtained automobile insurance data from the China Association of Automobile Manufacturers, including information on different types of new energy vehicles (pure electric and plug-in hybrid) and different functions (private cars, taxis, etc.). The high-resolution power outage data is obtained by web-scraping the power failure data published on the official website of each city government every day, including the power failure frequency and power failure duration of each district in the city. The control variables are compiled from the China City Statistical Yearbook. The EV charging station data comes from the China Electric Vehicle Charging Infrastructure Promotion Alliance. After removing missing data, the final sample consists of 310 cities, including 140 northern and 170 southern cities.

Two important sources of variations enable us to identify the causal relationship between power outages and EV adoption. The first is the large cross-sectional variation in the number of hours with power outages across different regions and cities. The data show that in 2020, the average number of hours with a power outage reached 162 h per year per district, and the standard deviation of the average hours with a

power outage per year per district was 188.14 h. This large standard deviation indicates that despite the relatively low average nationwide power outage hours, some cities still experienced severe power outages, which could impact the decision to adopt EVs. In contrast, EV adoption decisions for those living in cities with fewer power outages would be less impacted by the low number of power outages.

Second, the data spanning from November 2019 to September 2021 cover the large temporal or longitudinal variation in power outages, due to the two large-scale power outages happening in China during the study period. For example, several provinces experienced a >20-fold increase in power outages in Dec 2020 and Jan 2021 due to extreme cold weather. There were also large and extended power outages in Sep 2021 due to coal supply issues and recovery from the demand side electricity consumption. The sudden drastic increase in power outages in these two time periods gave us longitudinal variation for causal identification so that this study can compare the EV sales of a given region between periods with fewer power outages versus those with extended power outages. Such large-scale and extended power outages (an average of 1253 h of power outages per month per district across all cities in the data and a standard deviation of 2836.59 h in these 3 months) in these two time periods could impact consumers' decisions to adopt EVs.

In order to show that consumers in China do pay enough attention to power outages despite the lower average number of hours with power outages in regular periods, this study shows the Baidu keywords search index trends between three keywords: power outage, smog and air pollution, and climate change. Supplementary Fig. 1 shows that consumers pay comparable or even more attention to power outages compared to smog and air pollution and much more compared to climate change in the study period. This indicates that power outages may indeed affect consumers' relevant decision-making.

The majority of the power outages in the data of this study are in the residential area, as shown in Supplementary Fig. 2. Thus, power outages in the dataset will affect residential consumers' EV charging. In addition, power outages in industrial areas can still affect EV charging since people driving EVs may need to charge their EVs at work. Also, the dataset includes not only private vehicles owned by households but also vehicles owned by companies and organizations. The latter will need to charge at industrial areas as well. This study checked another data source (National Big Data Alliance of New Energy Vehicles, NDANEV) and found that >60% of EVs charge more than seven times a week. On average, an EV is charged 9.17 times per week. From NDANEV, this study randomly selected the charging records of 8768 EVs and drew the density distribution of the weekly charging frequencies of these vehicles (Supplementary Fig. 3). These data show that EV drivers charge a significant amount of times per week, and thus, power outages will influence the purchase intention of potential buyers.

## Calculation of the hours of power outages within a district

In the webscraped power outage data, each line item records the hours of a power outage that happened in a specific location within a district. For a given date, this study added all hours of power outages that happened within that day for the district. For example, on May 8th, 2020, within the Meilie District in Sanming City of Fujian province, three locations had power outages, one with 11.5 h, one with 11 h, and the third one with 12.5 h. This study then added all three power outages together, which gives us 35 h of power outages on that day for the Meilie District. As a result, when a district faces frequent and severe power outages across the locations within the district, the hours of power outages per district per day can exceed 24 h per day. Since this study does not have the number of electricity accounts within a district, it does not divide the total hours of power outages by the number of electricity accounts. However, not dividing the total hours by the number of accounts will not bias the estimation results since this study

uses the city-level fixed effects that can control for the average number of electricity accounts per district within a city.

As another robustness check, this study ran the regression models using the per capita definition of power outage. In this per capita definition, this study divides the hours and number of power outages added across different locations within a district by the number of permanent residents within a district. This study uses the 2020 permanent resident data, available from the 2021 China City Statistical Year Book, for this analysis. The results are shown in Supplementary Table 20, and this study still finds a negative and statistically significant relationship between power outages and EV sales.

## Base regression model

$$ln(\# \ of \ EV_{it}) = \beta_0 + \beta_1 * Outage_{it-k} + \gamma lnGDP_{it} + \lambda_{im} + \theta_{iy} + \epsilon_{it} \quad (1)$$

where $\# \ of \ EV_{it}$ is the number of new EVs in city $i$ in month $t$. $Outage_{it-k}$ is the lagged power outage in city $i$. This study uses a 1-month lag and a 2-month lag in two separate models. This study uses the lag because it can take time for consumers to decide on whether to purchase an EV, and it takes time for a new EV to get car insurance. This study uses two different measurements of power outages. The first measurement is the average number of power outages per district of a city in a month; a city can have many districts. The second measurement is the average number of hours with power outages at the district level. Please see Eqs. (2) and (3) for how this study defines these two power outage measurements. This study also controls the per capita GDP of each city. $\theta_{iy}$ is the city-by-year fixed effects, which control for time-variant confounding variables at the city level such as the incentives and policies in place for adopting EVs, population, consumer environmental awareness, and area of paved roads. $\lambda_{im}$ is the city-by-month fixed effects, which control for seasonal factors at the city level that can influence demand for vehicles and driving behaviors such as climate, temperature (which influences battery performance), and business cycles. More details of how the fixed effects can control for the spatial and temporal confounding factors can be found in Supplementary Note 1.

The two power outage measurements are defined below:

$$Outage \ times = \sum_{d=1}^{s} \sum_{j=1}^{n} D_{dj}/n \quad (2)$$

$$Outage \ hours = \sum_{d=1}^{s} \sum_{j=1}^{n} H_{dj}/n \quad (3)$$

where $d$ indicates a day of a given month; $j$ indicates a district of a given city; $D_{dj}$ is the number of outages in district $j$ on day $d$; $H_{dj}$ is the total number of hours with power outages in district $j$ on day $d$; $s$ is the total number of days of the month; $n$ is the total number of districts of the city.

There is a potential positive relationship between power outages and EV adoption since vehicle-to-grid (V2G) technology can enable the batteries in EVs to power the houses during outages. However, during the study period, the V2G was only on a very early small experimental and demonstration scale[72]. Regular households cannot use EVs to power their houses yet. As a result, this potential positive relation should not confound the empirical estimation.

## Statistical test for heterogeneity analysis

To examine how the impact of power outages differs by vehicle type, region, and GDP levels. This study first runs the same regression (Eq. (1)) for each of these categories separately to generate Fig. 3. Then in order to test whether differences are statistically significant, this study runs the following regression models, Eq. (4) & (5), adding the

interaction terms.

$$ln(\# \ of \ EV_{it}) = \beta_0 + \beta_1 * Outage_{it-k} * South + \eta * Outage_{it-k} + \gamma lnGDP_{it} + \lambda_{im} + \theta_{iy} + \epsilon_{it} \tag{4}$$

$$ln(\# \ of \ EV_{it}) = \beta_0 + \beta_1 * Outage_{it-k} + \beta_2 * Outage_{it-k} * High \ GDP + \gamma lnGDP_{it} + \lambda_{im} + \theta_{iy} + \epsilon_{it} \tag{5}$$

In the equation, *South* is an indicator variable and is equal to one if a city is located in the south and zero otherwise. The southern provinces are located to the south of the Huai river. *High GDP* is an indicator variable that is equal to one if a city is located in provinces with a per capita annual GDP >60,000 RMB.

## Robustness checks
This study conducts the following sets of additional analyses to test for the robustness of the main results.

Placebo test on non-EV vehicles: This study tests whether power outages negatively impact non-EV vehicles using Eq. (6). If not, this helps justify the main results that the negative impact on EVs is indeed due to power outages.

$$ln(\# \ of \ non\_EV_{it}) = \beta_0 + \beta_1 * Outage_{it-k} + \gamma lnGDP_{it} + \lambda_{im} + \theta_{iy} + \epsilon_{it} \tag{6}$$

Studies have shown that the majority (about 90%) of China's gas stations have backup power generation that provides enough electricity to power the gas pumps during a power outage[73,74] Despite the fact that gas pumps rely on electricity to function, power outages won't usually prevent conventional internal combustion engine vehicles from adding gasoline and thus won't negatively impact their purchases.

Different time lags: This study uses different time lags of power outages in the model. When zero lag is included, there is no statistically significant impact of power outages on the EV adoption of the same month (Supplementary Table 8). This is because it takes time for consumers to purchase an EV and register the car with insurance. One-month and 2-month lags have a statistically significant impact, indicating there are one- to 2-month lagged impacts. When using a 3-month lag, the effect is no longer statistically significant (Supplementary Table 8).

Add EV charging station as a control variable: The availability of EV charging stations can also impact EV adoption and could be correlated with lower power infrastructure (Supplementary Table 2). This study runs the following regression using Eq. (7):

$$ln(\# \ of \ EV_{it}) = \beta_0 + \beta_1 * Outage_{it-k} + \delta * Charging \ station_{it} + \gamma lnGDP_{it} + \lambda_{im} + \theta_{iy} + \epsilon_{it} \tag{7}$$

Instrumental variable approach: To address the endogeneity biases (reverse causality and missing variables), this study uses monthly extreme temperature days (daily maximum temperature > 89.6 °F or daily minimum temperature <32 °F) as an instrumental variable (IV) for power outages. Temperature is a valid instrument for the following reasons: (1) The short-run temperature (monthly variation) itself is exogenous and random. Short-run temperature is only influenced by natural factors. (2) Short-run temperature fluctuations should not directly influence consumers' intention to purchase EVs, thus satisfying the exclusion restriction condition. (3) Weather fluctuations can directly impact the electricity demand needed for space cooling and heating, and extreme weather can also impact the electricity supply.

This study conducts the first-stage regression before running Eq. (1) as

$$Outage_{it} = \alpha_0 + \alpha_1 * DD_{it} + \gamma lnGDP_{it} + \lambda_{im} + \theta_{iy} + \mu_{it} \tag{8}$$

where $DD_{it}$ indicates the monthly extreme temperature days variable. This study then uses the predicted values of power outages from Eq. (8) in the second-stage model when we run Eq. (1). As a result, short-run extreme temperatures can directly impact power outages, as confirmed by the first-stage results of the IV models (Supplementary Table 18 and Supplementary Table 19). The tables also report the weak instrument test results, and the Stock-Yogo weak ID test critical value and F statistics (all >10) show that the instruments are not weak.

## Different functional forms and model specifications
In order to choose the best functional form and to test for the robustness of the results, this study compares the results using the following four different functional forms via Eqs. (9)–(12):

$$Linear \ model: y = \beta_0 + \beta_1 x + \varepsilon_{it} \tag{9}$$

$$Semi\text{-}log \ model: ln \ y = \beta_0 + \beta_1 x + \varepsilon_{it} \tag{10}$$

$$Double\text{-}log \ model: ln \ y = \beta_0 + \beta_1 lnx + \varepsilon_{it} \tag{11}$$

$$Exponential \ model: y = \beta_0 + \beta_1 e^x + \varepsilon_{it} \tag{12}$$

This study hypothesizes that power outages have a negative impact on EV sales, and thus this study expects $\beta_1 < 0$ in all four models. In the linear model, the marginal impact of power outages on EV sales stays the same in all values of $x$ (power outage); in the semi-log model, the main model specification, the marginal impact becomes smaller as x becomes larger; in the double-log model, the marginal effect becomes smaller as $x$ becomes larger; in the exponential model, the marginal effect becomes larger when x becomes larger.

Results in Supplementary Table 9 show that, except for the linear model, all the other three models support a statistically significant negative relationship between EV sales and power outages. The coefficient of the power outage variable is not statistically significant in the linear model, indicating that the marginal effect of power outages on EV sales may not be constant in all values of power outages. In addition, since models may have specification errors, this study conducts the link test as proposed by Turkey[75] and Pregibon[76,77] to test for the model's effectiveness, namely to test whether the functional form that links the dependent and independent variables is correct. In particular, the link test requires the following steps using Eq. (13):

$$y = \alpha_0 + \alpha_1 \hat{y} + \alpha_2 \hat{y}^2 + error \tag{13}$$

where $\hat{y}$ is the predicted value of the dependent variable from each functional form. Then this study tests whether $\alpha_2$, the coefficient for the quadratic term $\hat{y}^2$, is zero, namely $H_0: \alpha_2 = 0$. If the null hypothesis is rejected, then the original functional form needs to be changed. The link test results are shown in Supplementary Table 10.

The results show that only the linear and semi-log (main method) models pass the link tests. However, since the linear model has an F-statistic that is not statistically significant, it indicates that the combination of independent variables does not influence y, and thus $\hat{y}^2$ does not explain y either. Thus $\alpha_2 = 0$ in the linear model does not imply that the linear functional form is correct. The null hypothesis in the double-log and exponential models are both rejected, implying that these two models may not be the best functional forms to model the relationship between power outages and EV sales.

To summarize, the results support that the semi-log model, the main model form, is the best specification to model the relationship between power outages and EV sales. In addition, this study has found some behavioral studies on habituation theory to support this functional form which has a larger marginal impact when the number of power outages is small. The habituation theory is formalized by several psychology studies[78–80]. The theory states that repeated presentation of a stimulus might decrease the response to the stimulus. In the context of this study, the marginal impact of power outages on consumers' EV purchase decision-making is larger initially. As the number of power outages increases, the room for EV sales reduction might be also shrinking, making the marginal impact smaller. Existing psychology laboratory experiments suggest that habituation is more ubiquitous than sensitization (the opposite of habituation where the repeated presentation of a stimulus causes an increase in the response to the stimulus) for most species[65], further supporting the choice of the semi-log functional form.

### Panel unit root test and cointegration test

This study uses the methods by Harris and Tzavalis[81] and Im et al.[82] to conduct the unit root tests of the panel data. The test results are shown in Supplementary Table 11. The results show almost all variables are stationary including new EV sales (increased number of EVs), GDP, and the number of power outages. One control variable, the cumulative number of charging stations, is a root process, but the unit root disappears after taking the first-order difference. The first-order difference measures the increase in charging stations, which is stationary.

Even though only one control variable (the cumulative number of charging stations) is non-stationary, this might still impact the model results. Thus, this study conducts a robustness check using the first-differenced number of all variables in the regression models. Results are shown in Supplementary Table 12. The model results are still consistent with those of the main models.

Cointegration analysis requires non-stationary variables to have the same order of integration. The panel unit root test results in Supplementary Table 11 show that EV sales, power outages, GDP, and the first difference of charging stations are all integrated of order zero. This study further conducts the panel co-integration test, namely using the following cointegration regression model via Eqs. (14) and (15):

$$\hat{Y}_t = \hat{\alpha}_0 + \hat{\alpha}_1 L2.O_t + \hat{\alpha}_2 X_t \qquad (14)$$

$$e_t = Y_t - \hat{Y}_t \qquad (15)$$

where this study tests for whether r $e_t$ is stationary; $\hat{Y}_t$ is the fitted value of EV sales (lnBHPEV), including both battery electric vehicles and plug-in hybrid electric vehicles; $O_t$ is the power outage times (outage); $X_t$ include the control variables. This study uses the two types of methods developed by Pedroni[83] and Kao[84] to test for the cointegration of panel data. The results are shown in Supplementary Table 13, which indicates that the variables have cointegration, with (0,0,0) order of cointegration relation. $e_t$ is stationary with unit root zero. This study also conducts the same cointegration tests for models using lnBEV and lnPHEV as the dependent variables as well as models using power outage hours (outageh) as the explanatory variable. The test results show the same conclusions for these other models.

### Potential simultaneity issue

The rapid increases in EV purchases will cause electricity consumption to rise. Such large-scale and unpredictable increases in consumption may increase the likelihood of power outages due to insufficient supply or grid deficiencies. If there is a simultaneous relationship between EV purchases and power outages, the single equation approach in the main model will bias estimates for the regression coefficients.

This study uses the method by Dumitrescu-Hurlin[85] to test whether there is a Granger causal relationship from EV purchases to power outages. Results in Supplementary Table 14 show that EV purchases do not cause changes in lagged power outages. In addition, we test for the relationship the other way around. Results in Supplementary Table 15 confirm the main conclusion that, indeed, lagged power outages cause the changes in EV sales. As a result, the simultaneity issue may not exist in the study period.

### Survey

A survey was conducted in Oct 2022. This study used a survey company called Credamo to conduct the survey. Credamo maintains a representative panel of survey respondents. This study asked the survey company to randomly select 1000 survey participants across China. In the end, there were 890 effective responses returned. The survey posed the question: Do you agree with the statement that extended power outages or an increasing frequency of power outages will reduce your willingness to purchase electric vehicles? Based on the results shown in Supplementary Fig. 4, 78.7% of the survey respondents agree with this statement, with 23% strongly agreeing. In addition, this paper incorporates many other factors such as charging infrastructure reliability, home charging concerns, public perception, income level, education level, road infrastructure access, government incentives, environmental awareness, urban congestion, access to public transport, and perception of low carbon technologies into the survey data. The regression results presented in Supplementary Table 25 indicate that power outages negatively affect the willingness of residents to purchase EVs. The survey results provide another piece of evidence that increasing power outages will influence the purchase intention of potential EV buyers. University of Maryland College Park (UMCP) Institutional Review Board (IRB) approved the survey study protocol. The IRBNet ID for this study is 2209723-1.

### Calculation of the carbon reduction benefits of EVs in China

Based on Caixin[46], in 2021, all new energy vehicles (NEVs) reduced carbon emissions by 15 million tons. In 2021, the number of new energy vehicles on the road was 7,840,000. Thus one clean energy vehicle can reduce carbon emissions by about 1500/784 = 1.91 ton/year. Based on Peng et al.[45], the social cost of carbon ranges from $ 20 –$ 200/ton of CO2. Thus on average, one clean vehicle in China can have a carbon emission benefit of 1.9 * (20 + 200)/2 = $ 210/vehicle/year.

According to Peng et al.[45], the upper bound of the economic benefits of combining health benefits and GHG emission reductions due to maximum penetration of alternate energy vehicles with high renewable electricity is equal to $ 1588 billion annually. The health benefits primarily come from the reductions in ambient $PM_{2.5}$ and $O_3$ concentrations. In 2020, the total number of vehicles on the road in China is about 260 million. When all are replaced by EVs, then on average one EV is responsible for $ 1588 billion/260 million = $ 6108/ vehicle/year. Then if the power outages double in a given year on average across China, the upper bound of decline in benefits from carbon emissions and health impacts is equal to $ 911 million/year U.S. dollars (10 * 0.99% * 405 * 310 * 6108 * 12).

### Reporting summary

Further information on research design is available in the Nature Portfolio Reporting Summary linked to this article.

## Data availability

The high-resolution geo-reference point level power outage data was obtained by scraping the daily power failure data published on the official website of each city government. The automobile insurance data was obtained from the China Association of Automobile Manufacturers. The EV charging station data comes from the China Electric Vehicle Charging Infrastructure Promotion Alliance. The EV charging

frequency data comes from the National Big Data Alliance of New Energy Vehicles. Other data used for this study are all retrieved from publicly available sources. All data used in this study are publicly available on GitHub (https://github.com/Deng0116/Power-outages-and-electrification) and Zenodo[88] (https://doi.org/10.5281/zenodo.12180607). Source data are provided with this paper.

## Code availability

All code used to process data and generate the figures, tables, and results in this study are publicly available on GitHub (https://github.com/Deng0116/Power-outages-and-electrification) and Zenodo (https://doi.org/10.5281/zenodo.12180607).

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

## Acknowledgements
This work was supported by the National Natural Science Foundation of China, received by Z.H.W. and B.W. (Reference Nos. 72243001, 72321002, 72141302, 72074026, 72222017, 72140002, 72104023); the Key Projects of the Philosophy and Social Sciences Program of the Ministry of Education, received by Z.H.W. (Reference No. 21JZD027); the National Funded Postdoctoral Program of China, received by N.N.D. (Reference Nos. GZC20233391, 2023M740236); and the Major Project of the National Social Science Foundation, received by B.W. (Reference No. 22&ZD104).

## Author contributions
Y.M.Q., B.W., and Z.H.W. designed the study; N.N.D, B.W., H.S., and J.L. completed data processing and visualization; N.N.D and B.W. completed econometric model related work; Z.H.W., Y.M.Q. and B.W. wrote the first draft; Z.H.W., B.W., N.H. and Y.M.Q. contributed to the interpretation of the results, N.N.D, Y.M.Q., X.C.S., N.H., and Y.D.W. contributed to the revision of the manuscript.

## Competing interests
The authors declare no competing interests.
