## [Peer Review File · Nature Communications]

REVIEWER COMMENTS

Reviewer #1 (Remarks to the Author):

This research builds and uses a structured dataset of power supply disruptions evidence to evaluate its effects on electric vehicle adoption in China's cities. For validation, a national survey has been conducted with 1000 observations with the following question: "Do you agree with the following statement: Extended power outages or increasing frequencies of power outages will reduce your willingness to purchase EVs". Authors also identify the following factors influencing the adoption of EVs: range anxiety, availability of EV charging stations, environmental awareness, and income. Among these factors, range anxiety and availability of charging stations indirectly imply that a steady supply of electricity to EVs is a critical factor. Authors also observed that there is a lack of empirical evidence on the impact of power supply disruptions on EV adoption.

Comments

Correlation does not imply causation/causality/connexion. I mention this because the authors have clearly identified some of the factors influencing the adoption of EVs, however, the study does not address those influencing factors. The survey results do not necessarily validate the correlation of one or more of those factors with EVs adoption. In this regard, the questionnaire might provide other insights (if asked during the survey) regarding other factors such as: charging infrastructure reliability, home charging concerns, public perception, income level (included in the survey but not in the analysis), education level (included in survey but not in the analysis), road infrastructure access, local/national government incentives, environmental awareness, urban congestion, access to public transport, perception of low carbon technologies, among others.

So, a major concern is relating the effect of outages on EV adoption without considering or assessing other factors/drivers. Another concern is that the data collection reflects the period of the pandemic COVID19. During this time, consumers were under lockdown which probably affected consumer choices. How did the research avoid any bias of the pandemic conditions over consumer choices?

Regarding academic writing style, I strongly recommend avoiding the use of "our" (134-word count) and "we" (86-word count) words. So, a proofreading/writing process would be necessary.

Reviewer #1 (Remarks on code availability):

The code "charging frequency_code.py" and "power outage area_code.R" are only about plotting. I am not familiar with .do and .dta file formats.

Reviewer #2 (Remarks to the Author):

This study sounds interesting in view of increasing temperature and adoption of EV vehicle in China.

I compliment for such study to authors.

Also the major limitation of this study is direct relation between number of power outages per district and the number of new EVs adopted per month. This is not valid argument.

I urge authors to include more socio-economical parameters and redo the analysis and submit the revised version. I will be happy to see it.

Reviewer #3 (Remarks to the Author):

This article is a thorough quantification of the impacts of power outages on EV adoption in China. It has the potential to be a highly cited work and could have great future work if applied to other markets. The conclusions are well supported by the methodology, data analysis, and interpretation with enough detail to be reproduced.

There are some minor edits that could improve the quality and impact of the paper:

References should not have a space before them and also should be placed after the period at the end of a sentence. Also, authors should change "include" in line 166 to "including."

An average of 21 hours per day with no power would prevent a full charge even at L2. Since this is the summation of hours that happened in all locations, not hours without power for an individual customer, I

suggest reporting an additional metric in this paragraph which might indicate what an individual would experience. Perhaps customer outage hours and then normalize by customer, or perhaps average hours without power for an individual customer day. This adjustment would improve reader perspective, so the additional description/metric is at the authors' discretion.

Given the extreme outage levels I think the use of "range anxiety" in line 173 is not quite accurate since it is not just anxiety, but an actual ability to charge that would prevent adoption.

From the legends on Figure 1 c and d, the captions for those figures is reversed.

In line 194, is this the number of EVs getting new insurance or simply buying insurance? The former would be a good proxy for new sales, the latter would be a proxy for total EV adoption.

In line 309 you say that this is based on the current fuel mix of the grid. Is this the fuel mix across China, or did you calculate

this based on city specific energy supplies?

In 377, you state that fleet adoption will not be sensitive to outages. Could outages change policies requiring adoption? The answer to this question is mostly just an idea for future work or discussion.

Below we respond to each referee comment. Referee comments are in blue, and our responses in plain text.

Reviewer #1 (Remarks to the Author):

This research builds and uses a structured dataset of power supply disruptions evidence to evaluate its effects on electric vehicle adoption in China's cities. For validation, a national survey has been conducted with 1000 observations with the following question: "Do you agree with the following statement: Extended power outages or increasing frequencies of power outages will reduce your willingness to purchase EVs". Authors also identify the following factors influencing the adoption of EVs: range anxiety, availability of EV charging stations, environmental awareness, and income. Among these factors, range anxiety and availability of charging stations indirectly imply that a steady supply of electricity to EVs is a critical factor. Authors also observed that there is a lack of empirical evidence on the impact of power supply disruptions on EV adoption.

Response: Thank you for your remarks on the overall work. We are grateful for your constructive feedback and the opportunity to improve our work.

Comments

Correlation does not imply causation/causality/connexion. I mention this because the authors have clearly identified some of the factors influencing the adoption of EVs, however, the study does not address those influencing factors. The survey results do not necessarily validate the correlation of one or more of those factors with EVs adoption. In this regard, the questionnaire might provide other insights (if asked during the survey) regarding other factors such as: charging infrastructure reliability, home charging concerns, public perception, income level (included in the survey but not in the analysis), education level (included in survey but not in the analysis), road infrastructure access, local/national government incentives, environmental awareness, urban congestion, access to public transport, perception of low carbon technologies, among others. So, a major concern is relating the effect of outages on EV adoption without considering or assessing other factors/drivers.

Response: We thank the reviewer for this important comment. To prove the causal relationship between power outages and EV adoption, we adopted a high-dimensional fixed effects model and Instrumental Variable (IV) estimation in the original manuscript. In addition, we incorporated more socio-economical variables as additional control variables to our original regression model and redid the analysis. Moreover, we appreciate your insight into the necessity of examining a broader range of factors beyond power outages that may affect consumer decisions regarding EV adoption. Therefore, we added the factors you mentioned to the questionnaire and re-conducted the analysis.

First, following the study of Tanaka and Okamoto (2021), we used high-dimensional fixed effects in the regression model. We included city-by-month fixed effects and city-by-year fixed effects in

the model. Controlling for city-varying time fixed effects in statistical models, involves incorporating dummy variables that account for specific effects unique to each city at each point in time (such as each year or each month). This method aims to capture and control for all unobserved heterogeneity that could influence the EV adoption, ensuring that the estimation of the relationship between the power outage and EV adoption is not biased by these unobserved factors. Specifically, the city-by-year fixed effects control for time-variant confounding variables at the city level such as the incentives and policies in place for adopting EVs, population, consumer environmental awareness, and area of paved roads, and the city-by-month fixed effects control for seasonal factors at the city level that can influence demand for vehicles and driving behaviors such as seasonal holiday, climate, temperature (which influences battery performance), and business cycles. These rich sets of fixed effects allow us to isolate the power outage effects on adoption of EV vehicle from many other factors.

Second, we conducted an IV approach, using monthly extreme temperature days as the instrumental variable to further address the endogeneity issues (reverse causality and omitted variable bias). IV estimation is a widely-used method used in statistics and econometrics to estimate causal relationships when randomized controlled experiments are not feasible and when there are concerns about endogeneity in observational data. In the context of our study, using extreme temperature days as an IV for power outages addresses potential endogeneity by leveraging natural variations in temperature that are presumably unrelated to other factors affecting EV adoption, such as consumer preferences or policy changes. Extreme temperatures can directly influence the frequency of power outages (due to increased demand on the grid or equipment failures) without directly affecting EV adoption decisions, except through their impact on power supply reliability. This setup allows for a clearer examination of how power outages influenced by extreme temperatures affect EV adoption, helping to establish a causal link between power reliability and EV adoption.

As suggested by the reviewer, in the revised manuscript we used more socio-economical variables as additional control variables to the regression model, including population, area of paved roads, coal gas supply, petroleum gas supply, educational level (measured by the number of teachers in ordinary higher education institutions), access to public transport (measured by the number of public buses) and charging infrastructure reliability (measured by the number of EV charging stations). These data all come from the China City Statistical Yearbook. To prevent these variables from being omitted, we used lower-dimensional fixed effects, including city fixed effect, year fixed effect, and month fixed effect, in the regression model. The regression results are shown in Table S22. We can see that the coefficients for the power outage variable remain significantly negative, confirming the robustness of the results of this study.

Table S22. Regression results with more socio-economic variables.

	lnNEV (BEV+PHEV) (1)	lnBEV (2)	lnPHEV (3)
L1. (outage times)	-0.0033*** (0.0007)	-0.0025*** (0.0007)	-0.0029*** (0.0007)
Socio-economic controls	YES	YES	YES

Constant	6.86*** (2.29)	6.09*** (2.51)	15.17*** (3.78)
Year FE	YES	YES	YES
Month FE	YES	YES	YES
City FE	YES	YES	YES
Number of observations	4842	4842	4842
Number of city	267	267	267
R-squared	0.92	0.92	0.78

Note: Standard errors in parentheses are clustered to city level. *P<0.1, **P<0.05, ***P<0.01. R-squared denotes the goodness-of-fit of the regressions. Socio-economic controls include per-capita GDP, population, area of paved roads, coal gas supply, petroleum gas supply, educational level (measured by the number of teachers in ordinary higher education institutions), access to public transport (measured by the number of public buses) and charging infrastructure reliability (measured by the number of EV charging stations).

In addition, regarding the survey data, we conducted another round of surveys during our manuscript revision (March 2024) and reorganized the questionnaire to incorporate the factors that you suggested, such as range anxiety and charging infrastructure reliability. The revised questionnaire utilized a 5-point Likert scale for both single and multiple-choice questions to capture respondents' attitudes and perceptions more accurately. To address the specific concern about the impact of power outages on EV adoption, we adopted two distinct approaches in our second-round survey, distributing 1000 randomly selected questionnaires for each approach. In the first set, we asked respondents about the level of power outages with the question "You often experience power outage in your living area" and provided options ranging from "strongly disagree" to "strongly agree". In the second set, we presented respondents with hypothetical scenarios of experiencing a certain number of power outages per week and asked about their likelihood of purchasing an EV under those conditions, also using a 5-point Likert scale. The regression results from both sets of questionnaires while controlling for the important factors that you mentioned that can impact EV adoption, as presented in Table S24 Columns 1 and 2, indicate that power outages negatively affect the willingness of residents to purchase new energy vehicles. These findings are consistent with our original analysis using actual sales data, thereby reinforcing the validity of our conclusions regarding the impact of power outages on EV adoption.

Table S24. The impact of power outage on EV purchase intention using survey data

	EV purchase intention (1)	EV purchase intention (2)
power outage level	-0.052** (0.022)	
power outage times		-0.054*** (0.011)
income level	0.083*** (0.016)	0.22*** (0.019)
education level	-0.048	-0.036

	(0.048)	(0.060)
charging infrastructure reliability	0.089***	0.55***
	(0.033)	(0.029)
home charging concern	-0.16***	-0.049***
	(0.017)	(0.025)
public perception	0.018	-0.0072
	(0.018)	(0.024)
road infrastructure access	0.055*	0.033
	(0.029)	(0.034)
government incentive	0.056**	0.11***
	(0.026)	(0.036)
environmental awareness	-0.018	0.053**
	(0.028)	(0.023)
urban congestion	-0.33***	0.0072
	(0.020)	(0.034)
access to public transport	0.018	0.026
	(0.024)	(0.029)
perception of low carbon technologies	0.0010	0.0048
	(0.027)	(0.026)
range anxiety	-0.16***	-0.15***
	(0.023)	(0.036)
City FE	YES	YES
Observations	976	856
R ²	0.80	0.72

Note: Standard errors in parentheses are clustered to individual level. *P<0.1, **P<0.05, ***P<0.01.

Another concern is that the data collection reflects the period of the pandemic COVID19. During this time, consumers were under lockdown which probably affected consumer choices. How did the research avoid any bias of the pandemic conditions over consumer choices?

Response: Thank you for your insightful comment regarding the potential impact of the COVID-19 pandemic on our research findings. We understand the importance of accounting for such unprecedented external factors in our analysis to ensure the robustness and validity of our conclusions. To address this concern, our study implemented two strategies to mitigate the potential bias caused by the pandemic conditions.

(1) Inclusion of a City Lockdown Dummy Variable: In the original manuscript, we incorporated a city lockdown dummy variable in our regression models to account for the immediate impact of COVID-19 lockdown measures on consumer behavior. This approach allowed us to control for the effects of lockdowns directly in our analysis. The results are consistent with the main conclusions and show that, conditional on the shocks of COVID restrictions power outages still have a statistically significant and negative impact on EV sales. Results also show that COVID lockdown policies had a negative impact on EV sales.

(2) Exclusion of Data from Cities Significantly Affected by COVID-19: As an additional measure, we excluded data from cities that were significantly impacted by COVID-19. We identified cities listed as medium or high-risk areas during the sample periods as regions severely affected by the pandemic. The data for the list of these medium and high-risk areas was sourced from State Council’s Government Affairs Platform of China. A total of 61 cities were listed as medium or high-risk areas during the sample periods. This step helped to further isolate the influence of pandemic conditions from the factors driving EV adoption. We provided the regression results without cities significantly affected by COVID-19 in Supplementary Information Table S23. The results are still consistent with the main conclusions.

Table S23. Regression results without cities significantly affected by COVID-19.

	lnNEV (BEV+PHEV) (1)	lnBEV (2)	lnPHEV (3)
L1. (outage times)	-0.0096*** (0.0015)	-0.0082*** (0.0014)	-0.012*** (0.0030)
lnGDP	-0.031 (0.029)	-0.014 (0.028)	-0.039 (0.040)
Constant	4.36*** (0.24)	4.05*** (0.23)	2.24*** (0.33)
Year*City FE	YES	YES	YES
Month*City FE	YES	YES	YES
Number of observations	3754	3754	3754
Number of city	241	241	241
R-squared	0.89	0.89	0.75

Note: Standard errors in parentheses are clustered to city level. *P< 0.1, **P< 0.05, ***P< 0.01. R-squared denotes the goodness-of-fit of the regressions. We excluded 61 cities that were listed as medium or high-risk areas during the pandemic.

Regarding academic writing style, I strongly recommend avoiding the use of “our” (134-word count) and “we” (86-word count) words. So, a proofreading/writing process would be necessary.

Response: Thanks for this detailed suggestion. In response to your comments, we have undertaken a thorough review of the manuscript to revise instances where “we” and “our” have been used inappropriately. We ensure that the revised manuscript adopts a more formal and objective tone by rephrasing sentences. For example, we changed the following sentence “*Our analysis demonstrates the importance of this factor for establishing policies...*” in the original manuscript into “*The analysis of this paper demonstrates the importance of this factor for establishing policies...*”.

Reviewer #1 (Remarks on code availability):

The code "charging frequency_code.py" and "power outage area_code.R" are only about plotting. I am not familiar with .do and .dta file formats.

Response: The code "charging frequency_code.py" is for plotting "*Figure S3. The density plot of the weekly charging frequency of EVs*" and the code "power outage area_code.R" is for plotting "*Figure S2. The distribution of power outages in residential versus industrial areas in our dataset*". These two figures are provided in the supplementary information.

The ".do" and ".dta" are file formats associated with Stata, a statistical software package commonly used for econometric analysis. ".do" files contain scripts and commands written in Stata's programming language, while ".dta" files store datasets in Stata format. The ".do" and ".dta" files are the code and data for the main regression results in the manuscript. You could use the Stata application to load the .dta file, then run the .do file to get the results.

We also offer more detailed instructions for ".do" and ".dta" files in the README section on the website <https://github.com/Deng0116/Power-outages-and-electrification>.

Reviewer #2 (Remarks to the Author):

This study sounds interesting in view of increasing temperature and adoption of EV vehicle in China. I compliment for such study to authors.

Response: We are grateful for the encouraging feedback. We have made every effort to improve the quality of our research based on your suggestions, and below are our point-by-point responses to each suggestion raised.

Also the major limitation of this study is direct relation between number of power outages per district and the number of new EVs adopted per month. This is not valid argument. I urge authors to include more socio-economical parameters and redo the analysis and submit the revised version. I will be happy to see it.

Response: Thanks again for your comments on the methodological advances. We agree that there are many socio-economical factors which could affect the adoption of EV vehicle. Therefore, in our regression model, we used high-dimensional fixed effects, allowing us to control for the impact of socio-economical factors on adoption of EV vehicle. Specifically, we adopted the city-by-year fixed effects to control for time-variant confounding variables at the city level such as the incentives and policies in place for adopting EVs, population, consumer environmental awareness, and area of paved roads, and the city-by-month fixed effects to control for seasonal factors at the city level that can influence demand for vehicles and driving behaviors such as climate, temperature (which influences battery performance), and business cycles. These rich sets of fixed effects allow us to isolate the power outage effects on adoption of EV vehicle from many socio-economical parameters.

We appreciate your suggestion to incorporate more socio-economic parameters into our analysis. We added more socio-economical variables to the model, including population, income level (measured by per capita GDP), area of paved roads, coal gas supply, petroleum gas supply, educational level (measured by the number of teachers in ordinary higher education institutions), access to public transport (measured by the number of public buses) and charging infrastructure reliability (measured by the number of EV charging stations). These data all come from the China City Statistical Yearbook. To prevent these variables from being omitted, we used lower-dimensional fixed effects in the regression model. The regression results are shown as the Table S22. We can see that the coefficients for the power outage variable remain significantly negative, confirming the robustness of the results of this study.

Table S22. Regression results with more socio-economical variables.

	lnNEV (BEV+PHEV) (1)	lnBEV (2)	lnPHEV (3)
L1. (outage times)	-0.0033*** (0.0007)	-0.0025*** (0.0007)	-0.0029*** (0.0007)
Socio-economical controls	YES	YES	YES

Constant	6.86 ^{***} (2.29)	6.09 ^{***} (2.51)	15.17 ^{***} (3.78)
Year FE	YES	YES	YES
Month FE	YES	YES	YES
City FE	YES	YES	YES
Number of observations	4842	4842	4842
Number of city	267	267	267
R-squared	0.92	0.92	0.78

Note: Standard errors in parentheses are clustered to city level. *P<0.1, **P<0.05, ***P<0.01. R-squared denotes the goodness-of-fit of the regressions. Socio-economical controls include population, per capita GDP, area of paved roads, coal gas supply, petroleum gas supply, number teachers in ordinary higher education institutions, number of public buses and the number of EV charging stations.

Reviewer #3 (Remarks to the Author):

This article is a thorough quantification of the impacts of power outages on EV adoption in China. It has the potential to be a highly cited work and could have great future work if applied to other markets. The conclusions are well supported by the methodology, data analysis, and interpretation with enough detail to be reproduced.

Response: We are grateful for the encouraging feedback. We have made every effort to improve the quality of our research based on your suggestions, and below are our point-by-point responses to each suggestion raised.

There are some minor edits that could improve the quality and impact of the paper: References should not have a space before them and also should be placed after the period at the end of a sentence. Also, authors should change "include" in line 166 to "including."

Response: We appreciate your attention to detail and fully agree with your suggestions. Regarding the formatting of references, we have carefully revised the manuscript to ensure that all references adhere to the correct formatting: no space before the references and placement after the period at the end of sentences. Additionally, we have amended the text in line 166 to change "include" to "including", as you correctly suggested.

An average of 21 hours per day with no power would prevent a full charge even at L2. Since this is the summation of hours that happened in all locations, not hours without power for an individual customer, I suggest reporting an additional metric in this paragraph which might indicate what an individual would experience. Perhaps customer outage hours and then normalize by customer, or perhaps average hours without power for an individual customer day. This adjustment would improve reader perspective, so the additional description/metric is at the authors' discretion.

Response: We thank the reviewer for this valuable comment. We presented the hours of power outages within a district for two reasons. First, the data on power outages made available by the government, which we utilized for our analysis, does not include specific information on the number of customers affected by each power outage within the districts. This limitation precludes us from calculating the average hours without power for an individual customer per day or normalizing outage hours by customer, as this would require detailed data on the number of customers experiencing each outage. Second, since our EV sales data is at the city level, we need to aggregate the power outage data to the city level in order to do the regression.

Additionally, we conducted another round of surveys during our manuscript revision (March 2024) to explore the impact of the extent of power outages experienced at the individual level on the willingness to purchase new energy vehicles. We adopted two distinct approaches in our second-round survey, distributing 1000 randomly selected questionnaires for each approach. In

the first set, we asked respondents about the level of power outages with the question “You often experience power outage in your living area” and provided options ranging from “strongly disagree” to “strongly agree”. In the second set, we presented respondents with hypothetical scenarios of experiencing a certain number of power outages per week and asked about their likelihood of purchasing an EV under those conditions, also using a 5-point Likert scale. The results are provided in Supplementary Information Table S24. The findings are consistent with the conclusions drawn from the analysis using power outage data and actual EV sales data at the city level in this paper.

Table S24. The impact of power outage on EV purchase intention using survey data

	EV purchase intention (1)	EV purchase intention (2)
power outage level	-0.052** (0.022)	
power outage times		-0.054*** (0.011)
income level	0.083*** (0.016)	0.22*** (0.019)
education level	-0.048 (0.048)	-0.036 (0.060)
charging infrastructure reliability	0.089*** (0.033)	0.55*** (0.029)
home charging concern	-0.16*** (0.017)	-0.049*** (0.025)
public perception	0.018 (0.018)	-0.0072 (0.024)
road infrastructure access	0.055* (0.029)	0.033 (0.034)
government incentive	0.056** (0.026)	0.11*** (0.036)
environmental awareness	-0.018 (0.028)	0.053** (0.023)
urban congestion	-0.33*** (0.020)	0.0072 (0.034)
access to public transport	0.018 (0.024)	0.026 (0.029)
perception of low carbon technologies	0.0010 (0.027)	0.0048 (0.026)
range anxiety	-0.16*** (0.023)	-0.15*** (0.036)
City FE	YES	YES
Observations	976	856
R ²	0.80	0.72

Note: Standard errors in parentheses are clustered to individual level. *P< 0.1, **P< 0.05,

*** $P < 0.01$.

Given the extreme outage levels I think the use of "range anxiety" in line 173 is not quite accurate since it is not just anxiety, but an actual ability to charge that would prevent adoption.

Response: We agree that our initial use of the term "range anxiety" may not fully capture the gravity of the charging challenges faced by potential EV buyers in these areas. In accordance with the reviewer's feedback, we have replaced "range anxiety" with "charging availability concerns" in the following sentence to accurately reflect that the primary concern is the actual capability to charge EVs.

"The lack of a steady electricity supply in these severely impacted cities could exacerbate charging availability concerns among potential EV buyers and thus lower EV adoption."

From the legends on Figure 1 c and d, the captions for those figures is reversed.

Response: Thanks for your detailed comments. We have corrected the captions to ensure they accurately match the content and data presented in Figures 1c and 1d.

In line 194, is this the number of EVs getting new insurance or simply buying insurance? The former would be a good proxy for new sales, the latter would be a proxy for total EV adoption.

Response: We use the number of EVs getting new insurance as a proxy for new EV sales, not simply buying insurance.

In line 309 you say that this is based on the current fuel mix of the grid. Is this the fuel mix across China, or did you calculate this based on city specific energy supplies?

Response: Thanks for the comment, a good point to clarify. In this study, our calculation for carbon reduction benefits is based on the fuel mix of the electric grid across China. Our decision to utilize a national fuel mix in our analysis stems from the unique structure and operational characteristics of China's electricity grid. Notably, China's grid is segmented into six major regional grids, facilitating the inter-provincial and even inter-regional dispatch of electricity resources. This means that the electricity supply to any given city is not solely dependent on locally generated power but is significantly influenced by the broader regional grid's capabilities and resources. In light of this, assessing the carbon reduction benefits based on the national or large regional energy mix provides a more accurate reflection of the actual electricity supply dynamics in China. It accounts for the interconnected nature of the grid and the country's energy strategy, which includes optimizing the distribution of electricity from areas of surplus to those of deficit, thereby smoothing out the variances in local energy mixes that would be encountered on a city-by-city basis.

In 377, you state that fleet adoption will not be sensitive to outages. Could outages change policies requiring adoption? The answer to this question is mostly just an idea for future work

or discussion.

Response: This is an insightful and important feedback. We agree that this is an important area for further investigation and discussion. We have revised our manuscript to include these considerations, thereby providing a more comprehensive view of the factors influencing electric FHV adoption and the resilience of related policies.

“There have been various policies and measures requiring the government to purchase and use electric vehicles for their official vehicles. Similarly, the central and local governments have stated in their policy documents that the shares of clean energy vehicles used for industries such as taxis, ride-hailing services, urban logistics and freight distribution, and express delivery need to be improved. As a result, there is policy pressure for companies and government agencies to increase the adoption of electric FHVs and thus their decisions will not be sensitive to outages. However, the effectiveness and sustainability of these policies may be challenged by infrastructure limitations, notably the reliability of the electricity supply. Given the critical role of a reliable electricity supply in the successful adoption of EVs, frequent or prolonged power outages could necessitate a re-evaluation of existing policies. Acknowledging and addressing these challenges through adaptive policies could pave the way for more resilient and effective electrification efforts in the transportation sector.”

Reference:

Tanaka, T., & Okamoto, S. (2021). Increase in suicide following an initial decline during the COVID-19 pandemic in Japan. *Nature human behaviour*, 5(2), 229-238.

REVIEWERS' COMMENTS

Reviewer #1 (Remarks to the Author):

I have reviewed the rebuttal and the manuscript. I recommend to accept the manuscript for publication.

Reviewer #2 (Remarks to the Author):

The execution of the manuscript seems logical and has a sound analysis. Methodology is clear in the manuscript. Critical analysis of the problem is well presented during the execution of the manuscript. Results of the manuscript seem correct technically. It is well written and technically sound and provides useful information on the issue.

Reviewer #3 (Remarks to the Author):

The authors have adequately addressed the comments on their previous version. The inclusion of additional survey results and analysis more closely tie the outage durations to the likelihood of purchasing an EV.